# Towards Expansive and Adaptive Hard Negative Mining: Graph Contrastive Learning via Subspace Preserving

## ABSTRACT

Graph Neural Networks (GNNs) have emerged as the predominant tool for analyzing graph data on the web and beyond. Contrastive learning (CL), a self-supervised paradigm, not only mitigates the reliance on annotations but also leads to breakthroughs in performance. The hard negative sampling strategy that benefits CL in other domains proves ineffective in the context of Graph Contrastive Learning (GCL) due to the message passing mechanism. Embracing the subspace hypothesis in clustering, we propose a method towards expansive and adaptive hard negative mining, referred to as Graph contRastive leArning via subsPace prEserving (GRAPE). Beyond homophily, we argue that false negatives are prevalent over an expansive range and exploring them confers benefits upon GCL. Diverging from existing neighbor-based methods, our method seeks to mine long-range hard negatives throughout subspace, where message passing is conceived as interactions between subspaces. Additionally, our method adaptively scales the hard negatives set through subspace preservation during training. In practice, we develop two schemes to enhance GCL that are pluggable into existing GCL frameworks. The underlying mechanisms are analyzed and the connections to related methods are investigated. Comprehensive experiments demonstrate that our method achieves state-of-the-art performance on multiple graph datasets and maintains competitiveness in various application settings. Our work contributes to the improvement of representation learning on web graphs, aligning with the scope of The Web Conference. Our code is available at https://anonymous.4open.science/r/Grape-code.

## CCS CONCEPTS

• **Information systems** → **Data mining**; • **Computing methodologies** → **Learning latent representations**; *Neural networks*; • **Mathematics of computing** → *Graph algorithms*.

## KEYWORDS

Graph neural networks, Graph contrastive learning, Hard negative mining, Subspace preserving, Web data mining

**ACM Reference Format:**

Anonymous Author(s). 2024. Towards Expansive and Adaptive Hard Negative Mining: Graph Contrastive Learning via Subspace Preserving. In *Proceedings of the Web Conference (WWW '24)*. ACM, New York, NY, USA, 14 pages. https://doi.org/XXXXXXX.XXXXXXX

## 1 INTRODUCTION

Graph data is ubiquitous in both real-world and virtual realms, encompassing a broad spectrum of areas such as social networks, molecular structures, trade circulation. Recently, GNNs have witnessed significant strides in the domain of analyzing graph data, exhibiting exceptional performance in tasks such as graph classification [90, 99], node clustering [16, 54], link prediction [98, 111] and graph generation [40, 93]. Following the pioneering contributions of GCN [32], GraphSAGE [20], GAT [71], etc., numerous GNN architectures have been developed and enhanced. Almost all GNNs are built upon the message passing mechanism between neighbors, where each node acquires feature information from its neighbors and contributes its own feature information. Analogous to most neural networks, GNNs are typically trained in a supervised manner and require an abundance of annotations.

Contrastive Learning (CL), as a category of self-supervised methods, has recently demonstrated a series of state-of-the-art performances in various domains [6, 7, 14, 101, 107]. These studies emphasize that the representations learned by CL perform comparably to supervised learning in downstream tasks. The essence of CL lies in learning representations that retain invariance under a variety of distortions, referred to as "data augmentations" [68, 69]. To achieve this, researchers develop InfoNCE objective [18, 53], which maximizes a lower bound of mutual information between augmented views [2, 25]. The core conception is to draw positive pairs closer while repelling negative pairs apart [19].

The breakthroughs of CL in computer vision have motivated studies to extend the analogous concepts from visual representation learning to graph data, referred to as Graph Contrastive Learning (GCL). These GCL methods achieve sota in both graph-level and node-level tasks [21, 67, 83, 97, 108, 109]. GCL adheres to the typical CL paradigm, albeit with specific variations [63, 72]. In general, the application paradigms of CL in visual, textual, and graph data domains can be illustrated as Figure 1. As demonstrated, existing research in GCL can be summarized into the following two main threads: (1) augmentation for graph [28, 37, 59, 65, 83, 94–96, 102, 106, 110], which aims to adapt semantic-preserving augmentation techniques from visual data to irregular graph data. (2) contrastive loss for graph [21, 38, 64, 84, 97, 102, 109], which explores the loss functions suitable for GNN training within CL framework. Our work falls into the latter category. Unlike other mainstream instance-discriminating backbones [9, 10, 22, 23, 70, 76] where instances do not exhibit explicit interactions, GNNs rely on message passing among neighbors. A notable issue arises where hard negative sampling techniques, proven to contribute in CL [5, 29, 36, 60, 82], does not confer benefits in GCL and may even impair performance, which has been discussed in [46, 64, 84, 108]. The main concept behind is that hard negatives in GCL are prone to being false negatives, consequently, pushing away the semantically similar representations leads to a degradation in performance.

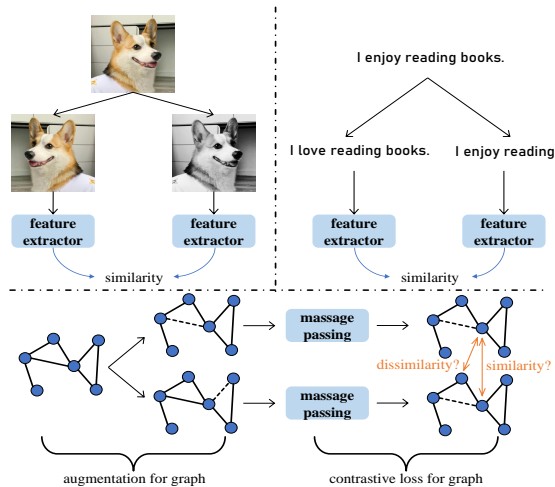

**Figure 1: A comparison of CL for visual, textual, and graph data. The irregularity of graph data and the message passing mechanism of GNNs distinguish GCL from CL in other domains. Graph convolutional operator introduces smoothing property among neighbors, while necessitating some technical changes to GCL.**

In this paper, we report that mining **expansive** and **adaptive** hard negatives enhances node-level tasks. To achieve both objectives, we introduce a negative hardness estimation scheme for GCL, aligning with the subspace preservation hypothesis in clustering. The core strength of our method lies in its ability to capture hard negatives beyond the scope of message passing and adjust the hard negatives set in a self-scaled manner. In node-level tasks, the concept of subspace preservation is intuitive. For instance, in a citation network, it can be elucidated as follows: from the semantic perspective, articles with the similar theme tends to share keywords (features); from the structural perspective, mutual citations within the same subfield are frequent whereas cross-domain article citations are limited. Prominent recommendation mechanisms within social or e-commerce networks, which curate personalized content for individual entities, have catalyzed the emergence of subspaces [58, 80, 81]. We provide theoretical and experimental analyses to illuminate why and how our method works. To the best of our knowledge, our work is the first to address the GCL through subspace techniques.

In summary, the main contributions of this paper can be encapsulate in threefold:

- We show that more expansive and adaptive hard negative mining is promising for enhancing node-level GCL. In line with this idea, we propose GRAPE, a novel negative hardness estimation method for GCL based on subspace theory.
- In GRAPE, the hard negatives beyond the scope of message passing can be captured and the hard negatives set can be adaptively scaled. Two schemes are devised to alleviate the influence of false negative samples on GCL. Besides, we provide a theoretical exposition of GRAPE's properties and its connection with related methods.
- In comparison to several advanced GCL methods, GRAPE exhibits superior performance on eight widely-used public

graph datasets. We conduct comprehensive experiments under various settings to thoroughly analyze the results and behaviors of GRAPE.

The proofs of involved theorems, experimental settings and supplementary experiments are relegated to the appendix.

## 2 RELATED WORK

In line with the focus of our work, we provide an overview of related works on graph contrastive learning and subspace preserving.

### 2.1 Graph Contrastive Learning

Amidst the increasing recognition of contrastive learning's expressive capability, DGI [72] and InfoGraph [63] first leverage the maximization of mutual information [25] at the node- and graph-level, respectively, to attain effective representations. In subsequent works, MVGRL [21] utilizes graph diffusion [15] to obtain augmented views and applies contrastive learning at both the node and graph levels. GMI [57] extends mutual information computations from vector spaces to the graph domain and assesses the correlation between input graphs and high-level hidden representations. GRACE [109], GCA [110] employ the InfoNCE-style objective and obtain node representations by treating others as negative samples, which serves as a baseline in follow-up research. To mitigate the sampling bias issue, BGRL [67] extends the BYOL [17] framework to graph. In this strand, CCA-SSG [97] optimizes a feature-level objective inspired by classical canonical correlation analysis. SpCo [43] is introduced as a spectral GCL module based on the general graph augmentation rule to enhance existing GCL methods. In another thread, ProGCL [85] estimates the probability of a true negative using a two-component beta mixture model. Empirical studies [108] verify that assigning higher weights to hard negatives or generating hard negatives fails to improve GCL. GDCL [104] jointly performs GCL and DEC [86]; nevertheless, this unsupervised process may lead to training collapse. COSTA [102] advocates generating covariance-preserving augmented features inspired by matrix sketching. HomoGCL [38] proposes utilizing the homophily in graph to filter positive pairs. PHASES [64] employs a progressive negative masking strategy to enhance tolerance between sample pairs. We recommend readers to refer to [45, 85, 87] for a comprehensive overview.

### 2.2 Subspace Preserving

One underlying tenet in machine learning is that the data contains certain type of structure for intelligent representation. From this, the subspace assumption, which runs through the research journey of machine learning, can be described as follows [41]: high-dimensional data is drawn from a union of multiple affine or linear subspaces. In a simplified perspective, affine subspace is more closely related to manifold learning [3, 24, 52, 61, 66, 91], whereas linear subspace aligns more closely with dictionary learning [27, 35, 49, 78, 79]. Over the past decade, subspace learning based on the self-expression model, which enjoys the benefits of the both, has made significant strides [12, 39, 41, 47]. The main divergence among these methods is the constraints imposed on the self-expression coefficients, such as sparse constraint [12, 55, 77], low-rank constraint [30, 41, 42], connectivity constraint [48, 89, 92],

and smooth constraint [4, 26, 34]. We adopt the fundamental principles of such methods to tackle hard negative mining in GCL. Both empirical investigations and theoretical analyses confirm the suitability in the context of GCL. Recent studies in graph dictionary learning [44, 74, 88] focus on sparse encoding for molecules, which are not directly related to our work.

# 3 METHODOLOGY

## 3.1 Notations and Preliminaries

Let $G = (A, X)$ denotes a graph with $n$ nodes, where $A \in \{0, 1\}^{n \times n}$ denotes the adjacency matrix and $X \in \mathbb{R}^{n \times d}$ denotes the feature matrix. Let $\hat{A} = A + I_n$ be the adjacency matrix with self-loops. The normalized adjacency matrix is then given by $\tilde{A} = D^{-\frac{1}{2}} \hat{A} D^{-\frac{1}{2}}$, where $D$ is the degree matrix. Vectors and matrices in this paper are denoted by bold lowercase and bold uppercase letters, respectively. Set $\{1, 2, \cdots, n\}$ is abbreviated by $[n]$ for simplicity.

Our objective is to train a GNN encoder $f_\Theta(A, X)$ in a label-scarce scenario, where $\Theta$ represents the network parameters. The output node embeddings are supposed to be directly applicable for downstream tasks, such as node classification and node clustering in this paper. Take GCN for example, the layer-wise forward-propagation operation at the $l$-th layer is formulated as:

$$Z^{(l)} = \sigma\left(\tilde{A} Z^{(l-1)} W^{(l)}\right), \qquad (1)$$

where $W^{(l)}$ is the trainable weights for feature transformation and $Z^{(l)}$ denotes the node embeddings at the $l$-th layer. Clearly, there is $Z^{(0)} = X$ at the initial layer. $\sigma(\cdot)$ denotes an activation function such as ReLU. In context of GCL, two views $G_1 = (A_1, X_1)$, $G_2 = (A_2, X_2)$ are generated by augmentation strategies [105] each epoch. $G_1$ and $G_2$ are fed into a Siamese GNN encoder [7] to produce node embeddings $\{u_i\}_{i=1}^n$ and $\{v_i\}_{i=1}^n$, respectively. The contrastive loss can then be computed, followed by backpropagation. The common baseline for graph contrastive loss is the InfoNCE-style loss in GRACE [109]. Specifically, the contrastive loss for $u_i$ is defined as:

$$\ell(u_i) =$$
$$-\log \frac{e^{\theta(u_i, v_i)/\tau}}{\underbrace{e^{\theta(u_i, v_i)/\tau}}_{\text{positive pair}} + \underbrace{\sum_{j \neq i} e^{\theta(u_i, v_j)/\tau}}_{\text{inter-view negative pairs}} + \underbrace{\sum_{j \neq i} e^{\theta(u_i, u_j)/\tau}}_{\text{intra-view negative pairs}}}, \qquad (2)$$

where $\theta(\cdot, \cdot)$ denotes cosine similarity and $\tau$ is the temperature parameter. The objective $\ell(v_i)$ is defined symmetrically. Then the overall loss is given as:

$$\mathcal{L} = \frac{1}{2n} \sum_{i=1}^n \left(\ell(u_i) + \ell(v_i)\right). \qquad (3)$$

It can be observed in Eq. (2) that by minimizing loss $\mathcal{L}$, embeddings of the same sample under two augmentations are pulled closer (**positives**), while embeddings of different samples are repelled away (**negatives**). For simplicity, in this paper, we represent $\ell(u_i)$ in the following form

$$\ell(u_i) = -\log \frac{\text{pos}(u_i)}{\text{pos}(u_i) + \text{neg}(u_i)}. \qquad (4)$$

## 3.2 Graph Contrastive Learning via Subspace Preserving

Recent studies [84, 108] report that considering all samples other than the anchor itself as negatives (Eq. (2)) unduly distances **false negatives** (i.e., samples of the same class as the anchor). This so-called "class collisions" phenomenon makes the marriage of CL and GNNs seem subtle and, as a result, leads to a performance decline. Hard negative mining provides a remedy to rectify this deficiency, where **hard negatives** refer to samples exhibiting a high degree of similarity to the anchor or having a greater likelihood of being false negatives. Let $\Phi_i$ be the hard negatives set of the $i$-th sample. With $\{\Phi_i\}_{i=1}^n$ identified, hard negative mining mainly employs two forms of loss: one explicitly treats $\Phi_i$ as positives [11, 38](referred to as "Positive" strategy), i.e., modifying the "pos" term in Eq. (4) to:

$$\text{pos}(u_i) = e^{\theta(u_i, v_i)/\tau} + \sum_{j \in \Phi_i} e^{\theta(u_i, v_j)/\tau} + \sum_{j \in \Phi_i} e^{\theta(u_i, u_j)/\tau}; \quad (5)$$

another strategy masks $\Phi_i$ within the negatives [8, 84] (referred to as "Mask" strategy), i.e., modifying the "neg" term in Eq. (4) to:

$$\text{neg}(u_i) = e^{\theta(u_i, v_i)/\tau} + \sum_{j \notin \Phi_i} e^{\theta(u_i, v_j)/\tau} + \sum_{j \notin \Phi_i} e^{\theta(u_i, u_j)/\tau}. \quad (6)$$

While both strategies are intuitive, their effectiveness on graphs remains to be thoroughly explored. We employ GRACE with two-layer GCN as a baseline and probe the quality of hard negative mining on three popular datasets. The results are shown in Table 1, with each value representing the average of 10 repeated runs. The different settings are explained as follows: "w/o MP" denotes GRACE without message passing, "$x$-hop" denotes selecting neighbors within $x$-hop as hard negatives, "$x$-hop$^*$" denotes selecting false negatives within $x$-hop as hard negatives, and "all labels" denotes selecting all false negatives as hard negatives. Here, 1-hop includes own neighbors for each node, while 2-hop encompasses the neighbors of neighbors and so forth. The "all labels" setting is an extreme scenario with all labels available in which $\ell(u_i)$ is akin to the tuplet loss in metric learning [31, 62].

**Table 1: Empirical study (node classification accuracy in percentage) on hard negative mining in GCL.**

| Datasets | Cora | | CiteSeer | | PubMed | |
|---|---|---|---|---|---|---|
| Settings | Positive | Mask | Positive | Mask | Positive | Mask |
| GRACE | 81.05 | | 71.27 | | 79.57 | |
| w/o MP | 44.34 | | 60.52 | | 72.94 | |
| 1-hop | 81.71 | 82.20 | 71.09 | 71.57 | 79.76 | 79.98 |
| 2-hop | 80.68 | 81.95 | 69.43 | 71.52 | 77.41 | 78.80 |
| 3-hop | 79.06 | 81.52 | 68.47 | 70.14 | 74.89 | 76.86 |
| 1-hop$^*$ | 82.49 | 81.83 | 71.42 | 71.06 | 80.54 | 80.31 |
| 2-hop$^*$ | 84.12 | 83.49 | 72.11 | 72.19 | 81.42 | 80.96 |
| 3-hop$^*$ | 86.61 | 86.15 | 74.86 | 73.52 | 83.26 | 83.20 |
| all labels | 97.57 | 94.63 | 95.93 | 93.44 | 97.01 | 95.52 |

Through empirical study, we make the following observations: (1) the performance of GCL significantly deteriorates in the absence of message passing; (2) "$x$-hop" settings provide limited benefits and may even be detrimental to GCL; (3) training under "$x$-hop$^*$" setting improves GCL performance; (4) "$x$-hop$^*$" setting with larger $x$ leads to more noticeable performance improvements. A follow-up query

arises as to whether it is feasible to narrow the gap between "x-hop" and "x-hop*"? Drawing insights from above observations: (2) and (3) inspire us to capture more "*precise*" hard negatives, while (4) encourages capturing "*expansive*" hard negatives. In self-supervised scenarios, the notion of "*precise*" may appear impractical, and thus, we pivot towards the pursuit of an "*adaptive*" solution.

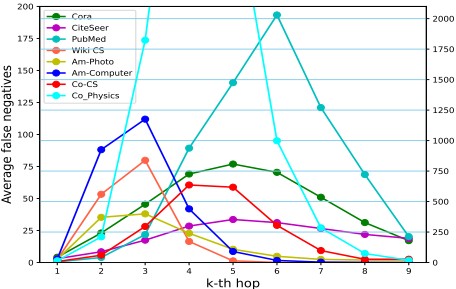

**Figure 2: The number of average false negatives at each hops on eight graph datasets.**

We plot the average distributions of false negatives on eight most commonly used network datasets in Figure 2. Cora and Citeseer correspond to the left coordinate axis, while the rest correspond to the right coordinate axis. It can be observed that false negatives are prevalent over an expansive range. This gives rise to the following concern: on the one hand, capturing more expansive false negatives approximates the performance under "all labels" setting; on the other hand, it is essential to prevent the capture of true negatives and thus avert the occurrence of 'x-hop' scenario. In other words, this is promising intuitively and entails practical risks.

For a specific anchor, its neighbors in close proximity frequently engage in message passing with it. Hence, its close neighbors are inherently hard to distance, whereas points with no message exchange with the anchor are susceptible to being pushed farther away, as shown in Figure 3a. This also explains why the "1-hop*" in Table 1 provides limited boost to the baseline. Beyond well-known graph homophily [50], we employ subspace preserving techniques to address this issue. The essence behind is to mine hard negatives across the entire subspace, rather than limiting it to graph-structured neighbors. Next, we provide the brief definition of subspace preserving.

DEFINITION 1. *(Subspace Preserving) The given data $\{x_i\}_{i=1}^{n}$ is drawn from a union of an unknown number $k$ of subspaces $\{\mathcal{S}_j\}_{j=1}^{k}$ with unknown dimensions $\{d_i\}_{i=1}^{k}$. $\mathcal{S}_j$ is subspace preserving if $\forall x_i \in \mathcal{S}_j$ can be expressed as a linear combination of other points in $\mathcal{S}_j$.*

Based on the so-called self-expressiveness property [12], the coefficients representing the contribution to the anchor can be obtained by solving the optimization problem:

$$\min_{c} \|z - Hc\|_2^2 + \lambda\Omega(c), \tag{7}$$

where $z \in \mathbb{R}^d$ is the representation of the anchor, and matrix $H \in \mathbb{R}^{d \times m}$ is formed by concatenating the representations of $m$ hard negatives of the anchor. $\Omega$ corresponds to a specific constraint on $c$. Note that the anchor in problem (7) represents any sample and we omit subscript $i$ for simplicity. Upon comparative analysis, we

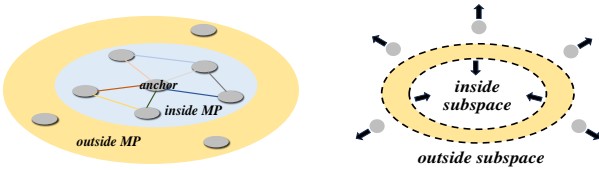

(a) Negatives for an anchor          (b) Classification on Cora

**Figure 3: Qualitative schematic of our method.**

opt for elastic net [112] as $\Omega$ in this paper, which is a combination of the $\ell_1$ and $\ell_2$-norms widely used in machine learning [13, 92, 103]. $\ell_1$-penalty encourages sparsity, while $\ell_2$-penalty promotes the connectivity. Furthermore, we expect to capture the consistent contribute from each hard negative throughout the entire process. The hard negatives selection for anchor $z$ turns out to be:

$$\min_{c} \sum_{l=1}^{L} \frac{1}{2d^{(l)}} \|z^{(l)} - H^{(l)}c\|_2^2 + \lambda\left(\mu\|c\|_1 + \frac{1-\mu}{2}\|c\|_2^2\right) \tag{8}$$

where $L$ is the number of network layers and $d^{(l)}$ is the dimension of the $l$-th layer. There is $z^{(l)} \in \mathbb{R}^{d^{(l)}}$ and $H^{(l)} \in \mathbb{R}^{d^{(l)} \times m}$. $\lambda > 0$ is the regularization parameter and $\mu \in [0, 1]$ controls the trade-off between two terms in the elastic net regularizer. As GNN performs message passing between neighbors at each layer, the subspaces at each layer may shift. Therefore, each forward propagation can be regarded as interactions between subspaces: some nodes are drawn into certain subspaces, while some are pushed out of their original subspaces. Scalar $\frac{1}{2d^{(l)}}$ is for scale equilibrium. The interpretation of the first term in Eq. (8) is to seek consistent coefficients $c$ across training layers. In other words, if a node consistently resides in the same subspace as the anchor, it is highly likely to be a false negative of the anchor. The magnitude of this possibility depends on the magnitude of self-expression coefficients.

Due to the sparse constraints, problem (8) can not be computed in closed form by SVD. Multiple solutions are provided below.
**Full parameterization**: If each position of $c$ is considered as a parameter, then problem (8) can be solved in fully parametric way, such as Iterative Shrinkage Thresholding Algorithm (ISTA) [1]. Moreover, $c$ can also be solved by gradient-based training. Since each sample serves as an anchor, the number of parameters in this strategy is $\sum_{i=1}^{n} m_i$. Updating these parameters during training may bring computational burdens on large-scale data.
**MLP parameterization**: In this scheme, self-expression coefficients can be computed on the lower-dimensional representations output from MLP. For example, SENet proposed in [100] employs a lightweight query and key network to parameterize the self-expression coefficients. Since MLP parameters does not depend on $n$, such methods alleviate computational overhead.
**Attentive parameterization**: Attentive models, such as GAT [71], presuppose varying contributions of distinct features. These models also utilize dimension-related memory to parameterize $c$.

The number of parameters in the above three ways decreases in order. Correspondingly, the expressive power decreases and the efficiency increases. Details are given in Appendix B. Since problem (8) is strongly convex, such accelerated proximal gradient mothed or linearized alternating direction method can be applied for seeking unique solution. Selecting non-zero indices in solution

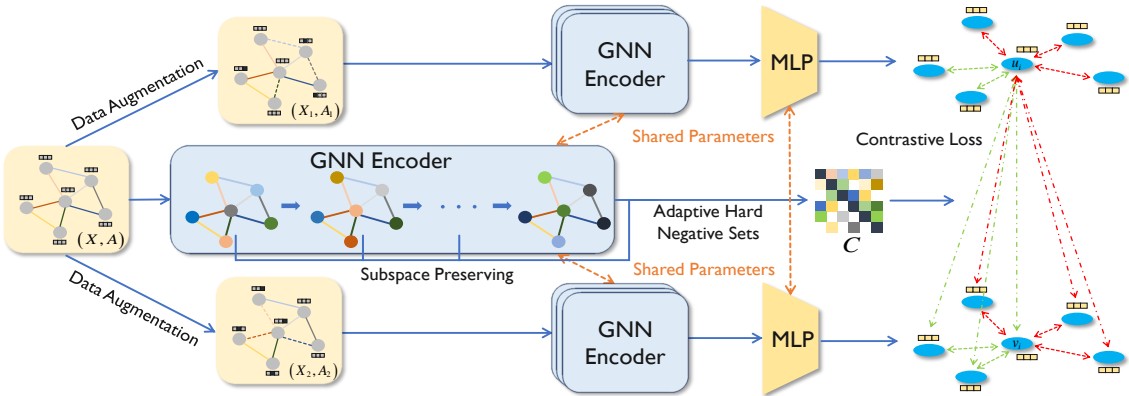

**Figure 4: The model architecture of GRAPE. The two views are generated through data augmentation of the initial graph. These three are fed into the parameter-sharing GNN encoder, where the projection header is alternative. The hard negatives set and the corresponding subspace coefficients $C$ are computed within the middle pathway. The green line in contrastive loss indicates hard negatives while the red line indicates true negatives, which are vary across epochs.**

$c$ and applying strategies like Eq. (5) and (6) may help attract false negatives within the subspace. Ideally, as shown in Figure 3b, true negatives are pushed farther away while false negatives are masked or explicitly drawn closer. It encourages the emergence of clear class boundaries, i.e. the golden band is stretched in Figure 3b. While the solution process is straightforward, problem (8) itself may not be static; in other words, the pre-selected matrix $H$ may not be optimal. This naturally prompts the question: how is the hard negatives set $H$ selected? Moreover, when dealing with large-scale data, it is extravagant to employ the self-representation of all samples on one single sample, we use a subset instead. Instead, we necessitate the adaptive selection of a subset.

Therefore, we aim to seek an adaptive matrix $H$ which can be self-scaled during the training process. The selected indices are expected to effectively preserve hard negative samples without becoming excessively large and causing training difficulties. Remark that problem (8) is independent for each sample. Next we introduce the definition of Adaptive Hard Negative Set for individual anchor.

DEFINITION 2. *(Adaptive Hard Negative Set) Assume $\tilde{c}(H)$ is the optimal solution of problem (8) with the i-th sample as the anchor. $\Phi$ is the adaptive hard negatives set of the i-th sample if the following conditions are satisfied:*

$$(a) \; \forall j \in \Phi, \; \tilde{c}^T\left(\left[H, z_j\right]\right) = \left[\tilde{c}^T(H), 0\right],$$
$$(b) \; \forall j \notin \Phi, \; \tilde{c}^T\left(\left[H, z_j\right]\right) = \left[q^T(H), \alpha_j\right], \quad (9)$$

*where $\left[q^T(H), \alpha_j\right]$ denotes the solution vector with scalar $\alpha_j \neq 0$.*

The interpretation of this definition is intuitive: $j$ within $\Phi$ make a contribution to the self-expression of the anchor (i.e., the optimal corresponding coefficient $\alpha_j$ are not zero), while $j$ outside $\Phi$ will not (i.e., the corresponding optimal coefficient equals to zero). In training, $\Phi$ can be ascertained via the following theorem.

THEOREM 1. *Assume $\tilde{c}(H)$ is the optimal solution of problem (8) with the i-th sample as the anchor. The auxiliary function is defined as*

$$g(j) = \sum_{l=1}^{L} \frac{1}{d^{(l)}} z_j^{(l)T} \left(z_i^{(l)} - H^{(l)} \tilde{c}(H)\right). \quad (10)$$

*Then hard negatives set can be computed by $\Phi = \{j \mid |g(j)| > \lambda\mu\}$.*

We can now give an understanding of what kind of samples are "hard" for a given anchor in the subspace framework. Theorem 1 implies that a sample is indispensable for subspace preserving if its representation sufficiently resembles the residual of existing self-expression. This diverges from homophily and similarity-based methods. Hence, our method exhibits "adaptive" in two aspects: On the one hand, as evident from the proof, it is clear that $c_j = 0$ is equivalent to $j \notin \Phi$. Therefore, it can be removed from the adaptive hard negatives set by updating $\Phi$ once. On the other hand, throughout the training process, updating $\Phi$ continuously expands the hard negatives set for the $i$-th sample. The distributional shifts from contrastive loss make it possible for $\Phi$ to capture long-range hard negetives.

Inspired by the OMP in dictionary coding [56], we aim for the gradual expansion of $\Phi$ with the training process. Beginning with an initial set, $\Phi$ can be periodically updated every few epochs to reduce additional time overhead while capturing expansive hard negatives. In addition, to avoid large-scale computations, the size of $\Phi$ can be controlled by confining hard negatives within a specified $K$-hop radius. The hyperparameter $K$ dictates the range of selectable hard negatives.

Combining the solutions of all subproblems, the self-expression matrix can be defined as $C = [c_1, \cdots, c_n]$, where $C_{ji}$ reflects the hardness of $j$ with respect to $i$. To incorporate the subspace information into the contrastive loss, the self-expression coefficients $C_{ij}$ is supposed to be mapped to the probability that $j$ serves as a false negative for $i$. This can be done through either a softmax operation or a linear mapping as follows:

$$(a) \; S_{ij} = \frac{\exp\left(|C_{ij}|/\sigma\right)}{\sum_{k \in \Phi_j} \exp\left(|C_{kj}|/\sigma\right)}, \quad (b) \; S_{ij} = \min\left\{\frac{|C_{ij}|}{\zeta}, \rho\right\}. \quad (11)$$

$S_{ij}$ in $(a)$ satisfies probabilistic properties and $\sigma$ is tunable. $S_{ij}$ in $(b)$ is proportionally scaled from $C_{ij}$, where $\zeta$ is the maximum value within a sampled subset $\{C_{ij}\}_{(i,j)}$. The truncated parameter $\rho$ controls the ceiling of $S_{ij}$ and is set to 1 by default. Thus, numerous values of $S_{ij}$ in equation $(b)$ can be equivalent to $\rho$. In turn, $S$ can be

symmetrized by $(S^T + S)/2$. Upon obtaining $S$, two schemes can be straightforwardly developed to enhance the performance of GCL. **GRAPE$_{mask}$:** GRACE in Eq. (2) treats all samples except itself as negatives, whose negatives set for anchor $i$ can be denoted as $\tilde{N}_i = [n] \setminus \{i\}$. While GRAPE estimates negatives' hardness and obtains the probability $S$ for false negatives in turn, it can subsequently excluded the highly probable false negatives from $\tilde{N}_i$. Specifically, in each epoch, $j$ is included in the false negatives set $\mathcal{F}_i$ for anchor $i$ with a probability of $S_{ji}$. The negatives set in this case turns out to be $N_i = \tilde{N}_i \setminus \mathcal{F}_i$. Therefore, the objective for $u_i$ in GRAPE$_{mask}$ is defined as:

$$\ell_{mask}(u_i) =$$
$$-\log \frac{e^{\theta(u_i, v_i)/\tau}}{e^{\theta(u_i, v_i)/\tau} + \sum_{j \in N_i}\left(e^{\theta(u_i, v_j)/\tau} + e^{\theta(u_i, u_j)/\tau}\right)}, \quad (12)$$

**GRAPE$_{pos}$:** GRACE in Eq. (2) exclusively treats itself as positives, whose positives set for anchor $i$ is $\tilde{\mathcal{P}}_i = \{i\}$. For anchor $i$, GRAPE$_{pos}$ incorporates $j$ into the positives set with a probability of $S_{ji}$ each epoch. The expanded positives set is denoted as $\mathcal{P}_i$. Therefore, the objective for $u_i$ in GRAPE$_{pos}$ is defined as:

$$\ell_{pos}(u_i) =$$
$$-\log \frac{e^{\theta(u_i, v_i)/\tau} + \sum_{k \in \mathcal{P}_i}\left(e^{\theta(u_i, v_k)/\tau} + e^{\theta(u_i, u_k)/\tau}\right)}{e^{\theta(u_i, v_i)/\tau} + \sum_{j \neq i}\left(e^{\theta(u_i, v_j)/\tau} + e^{\theta(u_i, u_j)/\tau}\right)}, \quad (13)$$

It is noteworthy that loss (13) is a variant of MIL-NCE [51]. Optimizing loss (13) enhances the overall similarity of positive pairs relative to negative pairs, rather than focusing on instance-specific distances.

Similar to GRACE, the overall contrastive loss is given as:

$$\mathcal{L}_{mask/pos} = \frac{1}{2n}\sum_{i=1}^{n}\left(\ell_{mask/pos}(u_i) + \ell_{mask/pos}(v_i)\right). \quad (14)$$

Reviewing the results in Table 1, we can empirically summarize that Grape$_{pos}$ are suitable for high-confidence false negatives, while Grape$_{mask}$ tolerates low-confidence false negatives. Therefore, Grape$_{mask}$ is deemed as a more robust scheme. The model architecture is presented in Figure 4 and the procedure for GRAPE is detailed in Appendix A.

## 3.3 Theoretical Analysis

**Why GRAPE works?** Reflecting on our motivation: we aim to identifies expansive and adaptive hard negatives as false negatives, which appears to be empirically derived. The essence behind this is the message-passing in GNNs: neighbors that encompass a substantial proportion of shared connections are not unduly distanced from each other. Therefore, local hard negative mining yield limited benefits. Recall the results in Table 1 that 1-hop* (even 2-hop*) does not significantly boost the baseline, while 3-hop* shows a leap, which interprets the pursuit of expansive hard negatives. Besides, the self-expression loss can be expanded as follows

$$\min_{c}\|z - Hc\|_2^2 \Leftrightarrow \min_{c} -2\sum_i z^T h_i c_i + \sum_{i,j} h_i^T h_j c_i c_j. \quad (15)$$

The first term endows larger self-expression coefficients for negatives similar to the anchor, while the second term endows smaller

coefficients for those highly similar to the other negatives. In GCL, the second term implies that the contributions of those involved in message passing with other hard negatives are diminished in self-expression, which is consistent with the intent in Figure 3a. This is rooted in its capacity to capture global long-range interactions, as discussed in [73]. Moreover, the regularizer in Eq. (8) exhibits sparsity as $\mu$ approaches 1 and group effect as $\mu$ approaches 0. It is worth noting that $\lambda$ and $\mu$ directly impact the tightness of hard negative selection. GRAPE with large values of $\lambda$ and $\mu$ results in a small hard negatives set.

By iteratively updating self-expression coefficients during training, the efficacy of GRAPE loss is qualitatively described as follows:

PROPOSITION 1. *In cases where GRAPE captures hard negatives $\{\Phi_i\}_{i=1}^n$ within each individual subspace, both $\mathcal{L}_{mask}$ and $\mathcal{L}_{pos}$ contribute to the inter-subspace separation and intra-subspace cohesion.*

Furthermore, GRAPE is associated with various methods, such as graph attention [71], nonlinear latent subspace clustering [55], and uniformity-tolerance dilemma [75], as revealed in Appendix D. **Maximizing mutual information** The improvement of GRAPE over the baseline can also be elucidated from the perspective of maximizing Mutual Information (MI):

THEOREM 2. *The contrastive loss in Eq. (14) gives a stricter lower bound of MI between input features $X$ and embeddings in two views $U$ and $V$, compared with the contrastive loss $\mathcal{L}$ in Eq. (3) proposed by GRACE. This can be written formally as*

$$-\mathcal{L} < -\mathcal{L}_{mask/pos} \leqslant \mathcal{I}(X; U, V) \quad (16)$$

Therefore, maximizing GRAPE loss corresponds to optimizing a more rigorous lower bound for the mutual information between node features and the acquired node representations, thereby furnishing a theoretical justification for the performance enhancement. **Complexity Analysis** The procedure for GRAPE is detailed in Appendix A. Compared to our baseline, GRACE, extra complexity arises from the periodic updating of hard negatives set $\{\Phi_i\}_{i=1}^n$ and the computation of self-expression coefficients $\{c_i\}_{i=1}^n$ every *intvl* epochs. Each of these $n$ independent subproblems can be solved concurrently in parallel. The additional time overhead is $O(Md)$, where $M$ represents the largest cardinality within $\{\Phi_i\}_{i=1}^n$. Since the hard negatives sets are restricted within the $K$-hop, there is $M \ll n$. Therefore, the additional time overhead is manageable.

## 4 EXPERIMENTS

## 4.1 Experimental Protocol

We conducted comparisons between GRAPE and ten advanced methods on eight node prediction datasets. The benchmark graph datasets include: **Cora**, **CiteSeer**, **PubMed**, **Wiki CS**, **Amazon Photo**, **Amazon Computers**, **Coauthor CS**, **Coauthor Physics**. They are all hosted by DGL package [1]. The dataset information is detailed in Appendix E.1. The comparative methods include: two supervised baselines (**GCN** [32], **GAT** [71]), two autoencoder-based baselines (**GAE** [33], **VGAE** [33]), eight state-of-the-art GCL methods (**DGI** [72], **GMI** [57], **MVGRL** [21], **GRACE** [109], **CCA-SSG** [97], **BGRL** [67], **ProGCL$_W$** [84], **COSTA$_{MV}$** [102]).

---

[1]https://https://github.com/dmlc/dgl

**Table 2: Node classification accuracy in percentage with standard deviation on eight real-world graph datasets. The bold and underlined values indicate the best and the runner-up results respectively.**

| Methods | Input | Cora | CiteSeer | PubMed | Wiki CS | Am-Photo | Am-Computer | Co-CS | Co-Physics |
|---|---|---|---|---|---|---|---|---|---|
| GCN | $X, A, Y$ | 81.32 ± 0.5 | 70.84 ± 0.7 | 77.69 ± 0.3 | 76.85 ± 0.1 | 92.16 ± 0.2 | 87.06 ± 0.5 | 92.54 ± 0.3 | 95.65 ± 0.2 |
| GAT | $X, A, Y$ | 82.57 ± 1.0 | 71.96 ± 1.0 | 77.51 ± 0.3 | 78.35 ± 0.1 | 91.45 ± 0.4 | 86.80 ± 0.3 | 91.98 ± 0.3 | 95.47 ± 0.2 |
| GAE | $X, A$ | 70.49 ± 1.8 | 63.56 ± 2.1 | 70.73 ± 1.0 | 72.08 ± 0.3 | 88.40 ± 0.3 | 82.93 ± 0.4 | 86.83 ± 0.6 | 92.50 ± 0.3 |
| VGAE | $X, A$ | 74.18 ± 1.1 | 64.85 ± 1.0 | 71.71 ± 0.5 | 73.49 ± 0.3 | 92.20 ± 0.1 | 86.37 ± 0.2 | 92.11 ± 0.1 | 94.52 ± 0.0 |
| DGI | $X, A$ | 82.90 ± 0.8 | 70.14 ± 0.8 | 76.80 ± 0.6 | 75.35 ± 0.1 | 91.61 ± 0.2 | 83.95 ± 0.5 | 92.15 ± 0.6 | 95.38 ± 0.1 |
| GMI | $X, A$ | 82.43 ± 0.9 | 69.85 ± 1.3 | 79.90 ± 0.2 | 74.85 ± 0.1 | 90.68 ± 0.2 | 82.21 ± 0.3 | OOM | OOM |
| MVGRL | $X, A$ | 83.20 ± 0.7 | 69.85 ± 1.5 | 78.28 ± 0.2 | 77.52 ± 0.1 | 91.74 ± 0.1 | 87.52 ± 0.1 | 92.11 ± 0.1 | 95.13 ± 0.0 |
| GRACE | $X, A$ | 81.05 ± 0.3 | 71.27 ± 0.4 | 79.57 ± 0.9 | 78.19 ± 0.0 | 92.15 ± 0.2 | 86.25 ± 0.3 | 92.26 ± 0.0 | 94.46 ± 0.6 |
| CCA-SSG | $X, A$ | 84.20 ± 0.4 | 72.57 ± 0.3 | 81.10 ± 0.2 | 78.42 ± 0.1 | 92.05 ± 0.3 | 87.95 ± 0.3 | 92.03 ± 0.1 | 95.40 ± 0.1 |
| BGRL | $X, A$ | 82.47 ± 0.2 | 71.13 ± 0.5 | 80.05 ± 0.2 | 78.06 ± 0.0 | 92.95 ± 0.3 | 88.19 ± 0.3 | **93.34 ± 0.1** | **95.54 ± 0.1** |
| ProGCL$_W$ | $X, A$ | 81.79 ± 0.6 | 68.63 ± 0.6 | 78.16 ± 0.2 | 78.30 ± 0.2 | 92.47 ± 0.2 | 87.23 ± 0.2 | 92.57 ± 0.1 | OOM |
| COSTA$_{MV}$ | $X, A$ | 81.66 ± 0.2 | 71.62 ± 0.4 | 78.39 ± 0.6 | 78.67 ± 0.1 | 92.20 ± 0.3 | 88.09 ± 0.0 | 92.96 ± 0.1 | 95.24 ± 0.0 |
| **GRAPE$_{mask}$** | $X, A$ | **85.18 ± 0.0** | 72.59 ± 0.0 | **81.50 ± 0.2** | **79.11 ± 0.1** | **93.32 ± 0.0** | **88.42 ± 0.1** | 92.78 ± 0.0 | 95.37 ± 0.0 |
| **GRAPE$_{pos}$** | $X, A$ | 85.07 ± 0.0 | **73.54 ± 0.1** | 79.84 ± 0.2 | 78.13 ± 0.1 | 92.95 ± 0.0 | 87.46 ± 0.1 | 92.29 ± 0.1 | 95.08 ± 0.0 |

For all augmentation-based methods, we adopt the most commonly used strategies for the graph augmentation: "edge removing" and "feature masking" [108]. At each epoch, "edge removal" randomly removes a certain proportion of edges from the original graph, while "feature masking" randomly masks a certain proportion of features. To be consistent with the comparison method, we configure the GNN encoder as a two-layer GCN. Self-supervised training is conducted on the entire graph and on the features of all samples. The embeddings obtained are fed into a semi-supervised linear classifier to get the final result. For Cora, CiteSeer, and PubMed datasets, we employ the standard split settings: 20 nodes per class are available for training, 500 nodes for validation and 1000 for testing. For the other datasets, we randomly assign 10% of the nodes for training, another 10% for validation, and allocate the remaining 80% for testing. The linear classifier is uniformly set to be a simple regularized logistic regression. The overall model is trained using the Adam optimizer.

We implement our GRAPE based on GRACE. The max training epoch is set to 100. The dimensions in the two-layer GNN encoder are set to 512 and 256, respectively. The learning rate for GRAPE is set to $1 \times 10^{-3}$, while that for linear classifiers is set to $1 \times 10^{-2}$. The interval for updating $C$ $intvl$ is fixed to 5 and the truncated parameter $\rho$ is fixed to 1. Our graph augmentation is achieved through a combination of 40% edge removal and 10% feature masking. The trade-off parameter $\lambda$ is selected within $\{10^{-1}, 10^{0}, 10^{1}, 10^{2}\}$ and $\mu$ is selected within $\{0, 0.1, \cdots, 0.9, 1.0\}$. The temperature parameter $\tau$ is selected within $\{0.1, 0.2, \cdots, 1.0\}$ and the range of hard negatives $K$ is selected within $\{1, \cdots, 5\}$. For all comparative methods, we adhere to the authors' default parameter settings and, where necessary, conduct parameter grid searches to achieve fair comparisons. Their implementations are all open-sourced. All experiments are conducted on NVIDIA RTX A6000 GPU with 48GB memory.

## 4.2 Main Results

The node classification results are presented in Table 2. The reported results are averaged over 10 runs with random seeds. The "Input" refers to data for training, where $X$, $A$ and $Y$ denotes feature matrix, adjacency matrix and label matrix respectively. OOM denotes out of memory. It can be observed that GRAPE achieves the state-of-the-art self-supervised performance on the first six

datasets and surpasses the performance of supervised baselines (GCN, GAT) on the first seven datasets. Compared to its baseline GRACE, GRAPE achieves a comprehensive improvement. The hyperparameters involved in the experiment are listed in Appendix E.2. We perform node clustering performance evaluations in the completely unsupervised case in Appendix F.1. In addition, we execute comparative experiments on heterophily graphs (where connections primarily occur between dissimilar nodes) and the results are presented in Appendix F.2. These results corroborate GRAPE's capacity for precise identification of false negatives.

## 4.3 How GRAPE Affects Training?

Subsequently, we conduct empirical investigations to explore what properties GRAPE learns and what hard negatives it captures. The experiments below are based on GRAPE$_{mask}$. Figure 5a illustrates the evolution of the percentage of false negatives within the hard negatives set $\{\Phi_i\}_{i=1}^{n}$ throughout the training process. Figure 5b depicts the distribution of hard negatives set over different hops after training. Despite the expansion of the hard negatives set, the proportion of false negatives within it scarcely declines. Besides, a substantial portion of the hard negatives set consists of large-hop neighbors. Both observations imply that we achieve expansive yet dependable sets of hard negatives through subspace preservation.

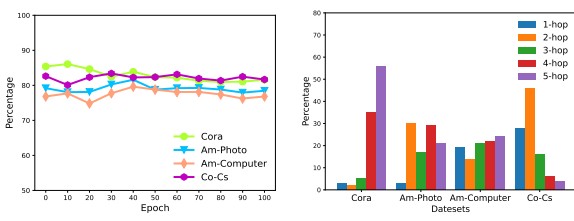

(a) False Negatives Percentage in $\Phi$    (b) Average Proportion at Hops

**Figure 5: Quality and distribution of hard negatives.**

Wang et al. [75] introduced the concept of uniformity-tolerance dilemma in contrastive representation. We employ the two metrics to showcase the difference between GRAPE and its baseline GRACE. Specifically, the cohesion (CO) and uniformity (UN) of the learned

embeddings can be defined as follows:

$$\text{CO} = \sum_{y_i = y_j} \left( f^T(x_i) f(x_j) \right), \ \text{UN} = \sum_{i,j} \exp \left( -f^T(x_i) f(x_j) \right) \quad (17)$$

$f$ denotes our GNN encoder $f_\Theta(A, X)$. A higher CO implies higher intra-class cohesion, while a higher UN implies a more uniform embedding distribution. The comparison of the two metrics for GRAPE and GRACE during training is depicted in Figure 6.

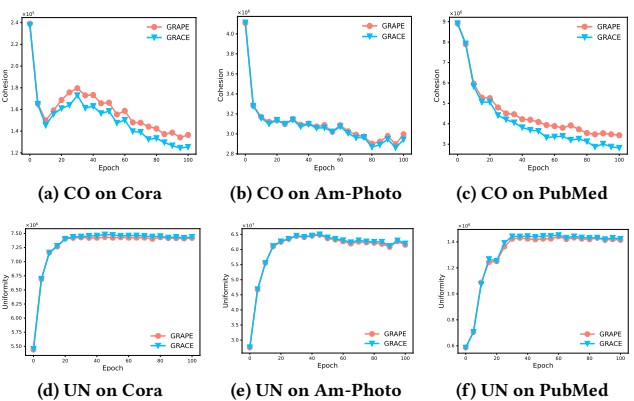

**(a) CO on Cora**    **(b) CO on Am-Photo**    **(c) CO on PubMed**

**(d) UN on Cora**    **(e) UN on Am-Photo**    **(f) UN on PubMed**

**Figure 6: Variation of cohesion and uniformity.**

At the beginning of training, CO and UN for both GRAPE and GRACE are nearly identical due to the similar initialization. As discussed ahead, GRAPE explicitly or implicitly brings the representations inside the same subspace closer, which strengthens the intra-class cohesion and reduces the global uniformity. With the expansion of the hard negatives set, the margin of cohesion between GRAPE and GRACE is enlarged, which is in line with our original intention. Since the mask mechanism of $\text{GRAPE}_{mask}$ is presented in a probabilistic form, uniformity doesn't exhibit significant decreases compared to GRACE. Additionally, Figure 7 shows how the test accuracy steadily improves as the GRAPE loss is optimized.

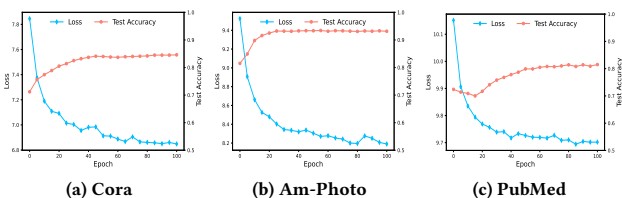

**(a) Cora**                **(b) Am-Photo**              **(c) PubMed**

**Figure 7: Variation of loss and test accuracy with training.**

Furthermore, the efficiency of GRAPE can be enhanced through aforementioned the scalable parameterization or by integration into negatives-independent methods. The corresponding results are provided in Appendix F.3 and F.4.

## 4.4 Visualization and Hyperparameter Study

In this subsection, we present intuitive results to illustrate the effectiveness of GRAPE. Figure ?? shows the distributions of the true/false negatives of the same anchor in different phases on Cora. The horizontal axis denotes the cosine similarity between negatives and anchor, which is non-negative due to the ReLU before output.

The variation with training is discernible, especially from (b) to (c), validating the efficacy of adaptive hard negative selection.

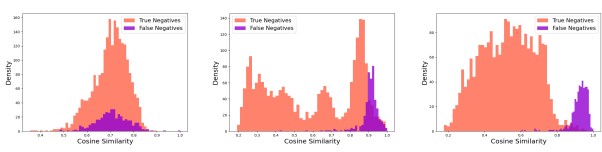

**(a) Initial (epoch=0)   (b) Training (epoch=50)   (c) Trained (epoch=100)**

**Figure 8: Negatives distributions in different phases.**

We present t-SNE visualization of GRAPE's running results without labels (i.e., before classification). As depicted in Figure 9, nodes are partitioned into multiple distinct clusters.

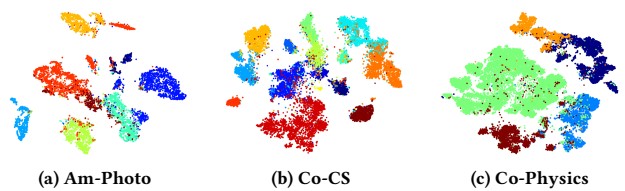

**(a) Am-Photo**          **(b) Co-CS**          **(c) Co-Physics**

**Figure 9: Visualization of node embedding without labels.**

The influence of the hyperparameters in GRAPE is examined to validate the feasibility. The sensitivity analysis of the two trade-off parameters in Eq. (8) is depicted in Figure 10. The test accuracy of GRAPE remains stable across a wide range of $\mu$ and $\lambda$, indicating its independence from meticulous parameter settings. Simultaneously, both parameters indeed exert an influence on the model. Besides, the parameter analyses of GRAPE under different interval $intvl$ and range $K$ are shown in Appendix F.5.

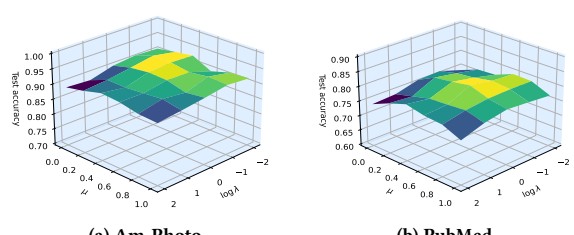

**(a) Am-Photo**                        **(b) PubMed**

**Figure 10: Sensitivity of trade-off parameters.**

## 5 CONCLUSION

In this paper we propose a novel method for estimating negatives' hardness in GCL. Our method emphasizes the potential in exploring expansive and adaptive negatives. These two goals are coupled in our subspace preserving scheme. We elucidate the motivation, provide empirical and theoretical underpinnings and conduct comprehensive experiments to dissect the effectiveness of GRAPE. Drawing from the contributions of this paper, we hopefully point out two interesting and promising avenues for further research. First, since subspace theory is not directly reliant on existing connections, it shows potential in addressing the impact of noisy, incomplete, or vulnerable graph structures on GNNs (a branch called graph structure learning). Second, self-expression contribute to preserving local structures and may serve as a form of constraint to slow down message passing for deeper GNNs.

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

## A PROCEDURE FOR GRAPE

---

**Algorithm 1** Procedure for GRAPE.

---

**Input:** Initial graph $G = (X, A)$, temperature parameter: $\tau$, trade-off parameters: $\lambda$, $\mu$, range of hard negatives: $K$, interval for updating $C$: $intvl$, maximum epochs: $Epo$.

1: **Initialization:** Randomly initialize the GNN parameters. Determine candidate set of hard negatives $\{\Phi_i\}_{i=1}$ and set the self-expression coefficients $\{c_i\}_{i=1}^n$ to zero.

2: **for** $epoch = 1$ to $Epo$ **do**

3:     Generate two augmented graphs $G_1 = (X_1, A_1)$ and $G_2 = (X_2, A_2)$. Feed $G, G_1, G_2$ into GNN encoder to obtain embeddings $\{z_i\}_{i=1}^n$, $\{u_i\}_{i=1}^n$ and $\{v_i\}_{i=1}^n$.

4:     **if** $epoch\%intvl == 0$ **then**

5:        Compute the self-expression coefficients $\{c_i\}_{i=1}^n$ on former hard negatives by solving Eq. (8).

6:        Update $\{\Phi_i\}_{i=1}^n$ within $K$-hop with solution $\{c_i\}_{i=1}^n$.

7:        Re-compute the self-expression coefficients $\{c_i\}_{i=1}^n$ on $\Phi$ and obtain $S$ by Eq. (11)

8:     **end if**

9:     Compute contrastive loss $\mathcal{L}_{mask/pos}$ in Eq. (12) or (13).

10:    Update $f_\Theta(A, X)$ with Adam by minimizing the overall loss in Eq. (14);

11: **end for**

**Output:** The trained $f_\Theta(A, X)$ and the node embeddings $\{z_i\}_{i=1}^n$.

---

## B LEARNING SELF-EXPRESSION COEFFICIENTS

The self-expression coefficients can be addressed through multiple efficient solutions. For instance, a common approach with iterative shrinkage thresholding algorithm (ISTA) [1] can be written as

$$c^{(t+1)} =$$

$$\Gamma_{\lambda\mu}\left(c^{(t)} + \varepsilon \sum_{l=1}^{L} \frac{1}{d^{(l)}}\left(H^{(l)^T}\left(z^{(l)} - H^{(l)}c^{(t)}\right) + \lambda(\mu-1)c^{(t)}\right)\right) \tag{1}$$

where $\varepsilon > 0$ is a step size and the soft-thresholding operator $\Gamma_\alpha : \mathbb{R}^n \to \mathbb{R}^n$ is defined by $\Gamma_\alpha(x) = (|x| - \alpha\mathbf{1})_+^T \text{sgn}(x)$. Since the problem is strongly convex, such accelerated proximal gradient mothed or linearized alternating direction method can be applied for seeking unique solution. The parameters to be solved are $\sum_{i=1}^n m_i$, related to the total number of nodes $n$.

In MLP-style parameterization, multi-layer mapping transforms the embeddings output by the GNN into another feature space. In the new feature space, $c_{ij}$ can be obtained through the dot product of the $i$-th and $j$-th samples, i.e., $c_{ij} = MLP_k(x_i)^T MLP_q(x_j)$ Readers may refer to [100] for implementation details.

Attentive parameterization can be implemented through various attention mechanisms, taking GAT as an example, where the self-expression coefficient can be parameterized as follows:

$$c_{ij} = \frac{\exp\left(\text{LeakyReLU}\left(\vec{a}^T\left[Wz_i\|Wz_j\right]\right)\right)}{\sum_{k \in h(i)} \exp\left(\text{LeakyReLU}\left(\vec{a}^T\left[Wz_i\|Wz_k\right]\right)\right)} \tag{2}$$

where $a \in \mathbb{R}^{2d_{(L)}}$ is the weight vector on feature and $\|$ is the concatenation operation. Both MLP parameterization and attentive parameterization have dimension-related parameters, making them suitable for large-scale graphs.

## C DETAILED PROOF

THEOREM 1. *Assume $\tilde{c}(H)$ is the optimal solution of problem (8). The auxiliary function is defined as*

$$g(h) = \sum_{l=1}^{L} \frac{1}{d^{(l)}} h^{(l)^T}\left(z^{(l)} - H^{(l)}\tilde{c}(H)\right). \tag{3}$$

*Then hard negatives set can be computed by $\Phi = \{h \mid |g(h)| > \lambda\mu\}$.*

PROOF. Problem (8) can be reformulated as

$$\min_c \frac{1}{2}\|z - Hc\|_2^2 + \lambda\left(\mu\|c\|_1 + \frac{1-\mu}{2}\|c\|_2^2\right) \tag{4}$$

where

$$z = \left[\frac{1}{d^{(1)}}z^{(1)^T}, \cdots, \frac{1}{d^{(L)}}z^{(L)^T}\right]^T$$
$$H = \left[\frac{1}{d^{(1)}}H^{(1)^T}, \cdots, \frac{1}{d^{(L)}}H^{(L)^T}\right]^T \tag{5}$$

By taking derivatives, the optimal solution $\tilde{c}(H)$ to problem (4) satisfies:

$$\lambda(1-\mu)\tilde{c}(H) = \Gamma_{\lambda\mu}\left(H^T\left(z - H\tilde{c}(H)\right)\right). \tag{6}$$

Let $\left[q^T(H), g(h)\right]^T$ be the optimal solution for problem

$$\min_c \|z - [H, h]c\|_2^2 + \lambda\left(\mu\|c\|_1 + \frac{1-\mu}{2}\|c\|_2^2\right). \tag{7}$$

Then there exist

$$\lambda(1-\mu)\left[q^T(H), g(h)\right]^T = \Gamma_{\lambda\mu}\left([H, h]^T\left(z - [H, h]\left[q^T(H), g(h)\right]^T\right)\right) \tag{8}$$

By splitting the counterpart terms, the following two equations hold:

$$\lambda(1-\mu)q(H) = \Gamma_{\lambda\mu}\left(H^T(z - Hq(H) - hg(h))\right) \tag{9}$$

$$\lambda(1-\mu)g(h) = \Gamma_{\lambda\mu}\left(h^T(z - Hq(H) - hg(h))\right) \tag{10}$$

If $h \notin \Phi$, then $\left[c^T(H), 0\right]^T$ is an optimal solution because it meets Eq. (9) and (10). Since the optimal solution to problem (7) is unique, condition a is thus satisfied. Since the optimal solution to problem (7) is unique, term (a) stipulated in the definition of $\Phi$ holds.

In the case where $h \in \Phi$, we show that $g(h)$ is not equal to 0. If $g(h) = 0$, due to the uniqueness of the optimal solution in problem (4), Eq. (9) deduces $q(H) = \tilde{c}(H)$. However, the obtained solution $\left[c^T(H), 0\right]^T$ does not satisfy Eq. (10). Therefore $g(h) \neq 0$ holds. Combining the above discussion, $\Phi$ is the adaptive hard negatives set by Definition 2, which completes the proof. □

PROPOSITION 2. *If GRAPE captures hard negatives $\{\Phi_i\}_{i=1}^n$ within each individual subspace, both $\mathcal{L}_{mask}$ and $\mathcal{L}_{pos}$ contribute to the inter-subspace separation and intra-subspace cohesion.*

PROOF. Compared to GRACE, GRAPE$_{pos}$ explicitly brings hard negative samples within the same subspace closer while repelling negatives outside the subspace. Our focus then turns to GRAPE$_{mask}$. From gradient analysis, the ratio of the gradients of negatives to that of positives can be defined following [75]:

$$r(\boldsymbol{u}_i, \boldsymbol{v}_j) = \left| \frac{\partial \ell(\boldsymbol{u}_i)}{\partial \theta(\boldsymbol{u}_i, \boldsymbol{v}_j)} \right| / \left| \frac{\partial \ell(\boldsymbol{u}_i)}{\partial \theta(\boldsymbol{u}_i, \boldsymbol{v}_i)} \right|, \qquad (11)$$

representing the relative penalty on negatives. The ratio in GRACE and GRAPE can be derived as follows:

$$\text{GRACE:} \quad r_1(\boldsymbol{u}_i, \boldsymbol{v}_j) = \frac{e^{\theta(\boldsymbol{u}_i, \boldsymbol{v}_j)/\tau}}{\sum_{k \neq i} \left( e^{\theta(\boldsymbol{u}_i, \boldsymbol{v}_j)/\tau} + e^{\theta(\boldsymbol{u}_i, \boldsymbol{u}_j)/\tau} \right)}, \quad (12)$$

$$\text{GRAPE:} \quad r_2(\boldsymbol{u}_i, \boldsymbol{v}_j) = \frac{e^{\theta(\boldsymbol{u}_i, \boldsymbol{v}_j)/\tau}}{\sum_{k \in \mathcal{N}_i} \left( e^{\theta(\boldsymbol{u}_i, \boldsymbol{v}_j)/\tau} + e^{\theta(\boldsymbol{u}_i, \boldsymbol{u}_j)/\tau} \right)}. \quad (13)$$

Clearly there is $r_2(\boldsymbol{u}_i, \boldsymbol{v}_j) \geqslant r_1(\boldsymbol{u}_i, \boldsymbol{v}_j)$, which implies that GRAPE imposes a greater penalty on negative pairs that are not within the same subspace. Moreover, assume that $S_{ij} = S_{ji} = 1$, i.e., $\boldsymbol{v}_i$ is not in the denominator of $\ell(\boldsymbol{u}_j)$ and vice versa. In this case, $\theta(\boldsymbol{u}_i, \boldsymbol{v}_j)$ is not penalized explicitly. Apart from self-alignment, the subproblem involving $\boldsymbol{u}_i$ in the process of minimizing $\mathcal{L}_{mask}$ is equivalent to:

$$\min_{\boldsymbol{u}_i} \sum_{i \notin \mathcal{N}_k \vee k \notin \mathcal{N}_i} \left( e^{\theta(\boldsymbol{u}_i, \boldsymbol{u}_k)} + e^{\theta(\boldsymbol{u}_i, \boldsymbol{v}_k)} \right) \qquad (14)$$

If we consider the first-order Taylor expansion of the problem and omit the second or higher-order infinitesimal terms, problem (14) simplifies to

$$\min_{\boldsymbol{u}_i} \sum_{i \notin \mathcal{N}_k \vee k \notin \mathcal{N}_i} \left( \theta(\boldsymbol{u}_i, \boldsymbol{u}_k) + \theta(\boldsymbol{u}_i, \boldsymbol{v}_k) \right). \qquad (15)$$

It is clear that there is a unique solution to the above problem. If $i$ and $j$ belong to the same subspace, the overlap between set $\{k \mid i \notin \mathcal{N}_k \vee k \notin \mathcal{N}_i\}$ and set $\{k \mid j \notin \mathcal{N}_k \vee k \notin \mathcal{N}_j\}$ appears to be high. Hence, the optimal solutions of the subproblems for $\boldsymbol{u}_i$ and $\boldsymbol{v}_j$ tends to exhibit high similarity. As a result, $\boldsymbol{u}_i$ and $\boldsymbol{v}_j$ are implicitly drawn closer by updating the network parameters. Thereby we prove the Proposition 2 qualitatively.                              □

THEOREM 2. *The contrastive loss in Eq. (14) gives a stricter lower bound of MI between input features $X$ and embeddings in two views $U$ and $V$, compared with the contrastive loss $\mathcal{L}$ in Eq. (3) proposed by GRACE. This can be written formally as*

$$-\mathcal{L} < -\mathcal{L}_{mask/pos} \leqslant \mathcal{I}(X; U, V) \qquad (16)$$

PROOF. We only consider the GRAPE$_{mask}$ scheme, while GRAPE$_{pos}$ can be analogized, since the proof is trivial. The contrastive loss in GRAPE$_{mask}$ and GRACE can be reformulated as follows:

$$\ell_{mask}(\boldsymbol{u}_i) = -\log \frac{e^{\theta(\boldsymbol{u}_i, \boldsymbol{v}_i)/\tau}}{e^{\theta(\boldsymbol{u}_i, \boldsymbol{v}_i)/\tau} + \sum_{j \in \mathcal{N}_i} \left( e^{\theta(\boldsymbol{u}_i, \boldsymbol{v}_j)/\tau} + e^{\theta(\boldsymbol{u}_i, \boldsymbol{u}_j)/\tau} \right)},$$
$$(17)$$

$$\ell(\boldsymbol{u}_i) = -\log \frac{e^{\theta(\boldsymbol{u}_i, \boldsymbol{v}_i)/\tau}}{e^{\theta(\boldsymbol{u}_i, \boldsymbol{v}_i)/\tau} + \sum_{j \neq i} \left( e^{\theta(\boldsymbol{u}_i, \boldsymbol{v}_j)/\tau} + e^{\theta(\boldsymbol{u}_i, \boldsymbol{u}_j)/\tau} \right)}, \quad (18)$$

It's clear that the overall loss satisfies $\mathcal{L} > \mathcal{L}_{mask}$.

The InfoNCE loss [18, 53] can be formulated as

$$\mathcal{L}_{NCE} = -\mathbb{E}_{p(\boldsymbol{u}, \boldsymbol{v})} \left( \frac{1}{n} \sum_{i=1}^n \log \frac{e^{\theta(\boldsymbol{u}_i, \boldsymbol{v}_i)/\tau}}{\frac{1}{n} \sum_{j=1}^n e^{\theta(\boldsymbol{u}_i, \boldsymbol{v}_j)/\tau}} \right). \qquad (19)$$

Analogously, replacing the empirical estimation by the expectation, the contrastive loss in GRAPE can be reformulated as

$$\mathcal{L}_{mask} = -\mathbb{E}_{p(\boldsymbol{u}, \boldsymbol{v})} \left( \frac{1}{2} \left( \ell_{mask}(\boldsymbol{u}) + \ell_{mask}(\boldsymbol{v}) \right) \right), \qquad (20)$$

where

$$\ell_{mask}(\boldsymbol{u})$$
$$= -\log \frac{e^{\theta(\boldsymbol{u}, \boldsymbol{v})/\tau}}{e^{\theta(\boldsymbol{u}, \boldsymbol{v})/\tau} + \sum_{j \in \mathcal{N}_u} \left( e^{\theta(\boldsymbol{u}, \boldsymbol{v})/\tau} + e^{\theta(\boldsymbol{u}, \boldsymbol{u})/\tau} \right)}$$
$$= -\frac{1}{n} \sum_{i=1}^n \log \frac{e^{\theta(\boldsymbol{u}, \boldsymbol{v})/\tau}}{\frac{1}{n} \sum_{j=1}^n e^{\theta(\boldsymbol{u}, \boldsymbol{v})/\tau} + \sum_{j \in \mathcal{N}_u} \left( e^{\theta(\boldsymbol{u}, \boldsymbol{v})/\tau} + e^{\theta(\boldsymbol{u}, \boldsymbol{u})/\tau} \right)}.$$
$$(21)$$

This clearly gives $\mathcal{L}_{NCE} \leqslant \mathcal{L}_{mask}$. The InfoNCE estimator is a lower bound of the true MI, i.e., $-\mathcal{L}_{NCE} \leqslant \mathcal{I}(U; V)$. From the data processing inequality in information theory, it can be deduced that $\mathcal{I}(U; V) \leqslant \mathcal{I}(X; U, V)$. Collecting the above discussion, there exists $-\mathcal{L}_{mask} \leqslant \mathcal{I}(X; U, V)$, which completes the proof.      □

# D   RELATED METHODS

**Graph attention** The connection between self-expression and attention mechanism has been deliberated in [73]. In this context, we underscore two pivotal distinctions between GRAPE and existing graph attention methods, such as GAT [71] and its variants: (1) Irrespective of the training paradigm, it's essential to note that in graph attention, the attention coefficients are learned with respect to the loss, whereas in GRAPE, the self-expression coefficients $C_{ij}$ adapt during the training process and exert an influence on the loss function. (2) Diverging from the structure-dependent attention coefficients in graph attention methods, GRAPE's self-expression coefficients are not inherently contingent on the graph structure. This feature expands the application domain of GRAPE to noisy graphs, corrupted graphs, or even heterophily graph structures.

**Nonlinear latent subspace clusterng** The fundamental concept underlying latent subspace clustering [55] and its variants is that the subspaces delineating the representation of raw data may not be readily discernible, hence, they may be partitioned into subspaces via learnable transformations. The features of the samples may not lie within $k$ specific subspaces, but their intrinsic semantics are encapsulated within $k$ larger subspaces. GRAPE also enjoys this concept. Each layer of the GNN encoder serves as a nonlinear transformation of the previous layer, wherein point pairs that preserve the same subspace structure are included into the hard negatives set.

**Uniformity-tolerance dilemma** In [75], 'tolerance' pertains to the degree of similarity between false negatives, while 'uniformity' signifies the separability of all negatives. These two concepts are somewhat contradictory, a phenomenon commonly referred to as the uniformity-tolerance dilemma. Though temperature-based approaches [75] are straightforward, temperature $\tau$ tend to be global and challenging to learn. An intuitive comparison, as depicted in Figure 11, showcases the average distribution of negatives on Cora.

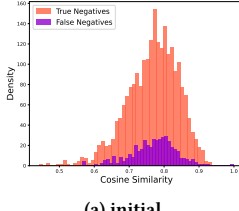 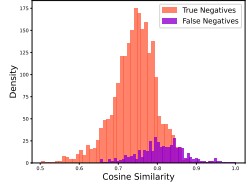 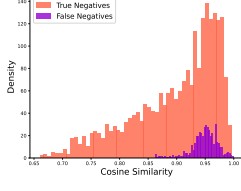 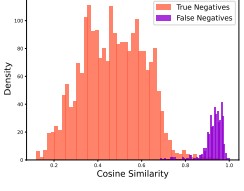 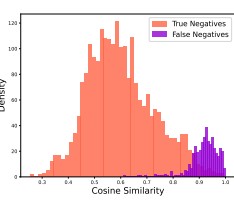

| (a) initial | (b) with high uniformity | (c) with high tolerance | (d) label available | (e) GRAPE |

Figure 11: Similarity histograms of negatives

It becomes apparent that solely adjusting the temperature parameter $\tau$ does not suffice to achieve local tolerance, whereas GRAPE can effectively approximate the distribution in supervised scenario. The setting with high uniformity (small $\tau$) pushes false negatives further away, whereas the setting with high tolerance (large $\tau$) makes it challenging to discriminate true negatives. Compared to temperature-based methods, GRAPE alleviates class collisions effectively. [64] demonstrates how hard negative masking enhances local tolerance, therefore, GRAPE can be regarded as a tolerance-preferred method.

## E EXPERIMENTAL SETUP

### E.1 Datasets

**Cora**, **CiteSeer**, and **PubMed** are three prominent citation networks, where features correspond to keywords, directed edges represent citation relationships, and labels denote subject.

**Wiki CS** is a database comprising nodes corresponding to Computer Science articles, where edges base on hyperlinks, features base on the average of pretrained GloVe word embeddings, and classes represent various branches within the field.

**Amazon-Photo** and **Amazon-Computers** are segments within the Amazon co-purchase graph. In these segments, nodes correspond to goods, edges denote frequent co-purchases between goods, node features are derived from bag-of-words encoded product reviews, and class labels are determined by the product categories.

**Coauthor-CS** and **Coauthor-Physics** are co-authorship graphs obtained from the KDD Cup dataset. In these graphs, nodes correspond to authors, edges represent collaborative research relationships, node features encapsulate paper keywords associated with each author's publications, and class labels denote the most active research fields for each author.

Statistics of datasets are shown in Table 3 and the pytorch usage of these datasets is detailed in document [2].

### E.2 Hyperparameter Setting

Table 3 lists the hyperparameters for our main performance experiments. The two-layer GCN maintains its dimensions at 512 and 256 throughout. Notably, the parameters do not require meticulous tuning, implying that GRAPE obviates the need for parameter search. This indicates the sound scalability of our method.

## F SUPPLEMENTARY EXPERIMENTS

### F.1 Node Clustering

The evaluation of node clustering is similar to classification, except that k-means is employed for clustering. Clustering performance is

---

2https://docs.dgl.ai/api/python/dgl.data.html

assessed using NMI and ARI, where higher values of these metrics indicate superior clustering results. The average results of the 5 runs are presented in Table 4.

**Table 4: Node clustering results in percentage on three graph datasets.**

| Datasets | PubMed | | Am-Photo | | Am-Computer | |
|---|---|---|---|---|---|---|
| Metrics | NMI | ARI | NMI | ARI | NMI | ARI |
| GAE | 24.41 | 24.35 | 57.30 | 49.45 | 42.80 | 24.68 |
| VGAE | 21.44 | 18.54 | 54.18 | 40.25 | 42.88 | 23.74 |
| DGI | 27.96 | 29.50 | 44.77 | 35.11 | 37.35 | 20.25 |
| GMI | 24.96 | 25.04 | 50.47 | 42.22 | 45.88 | 30.50 |
| MVGRL | 31.96 | 30.79 | 56.48 | 44.06 | 29.18 | 19.57 |
| GRACE | 26.01 | 28.44 | 61.93 | 50.41 | 48.76 | 33.85 |
| CCA-SSG | 25.04 | 28.22 | 62.30 | 53.87 | 49.62 | 36.64 |
| BGRL | 25.63 | 26.44 | 63.29 | 53.48 | 49.70 | 32.51 |
| ProGCL$_W$ | 27.26 | 29.50 | 60.54 | 48.32 | 43.29 | 28.44 |
| COSTA$_{MV}$ | 27.91 | 28.59 | 58.69 | 48.87 | 45.54 | 36.90 |
| **GRAPE$_{mask}$** | 32.13 | 31.80 | 65.33 | 57.72 | 53.06 | 38.49 |
| **GRAPE$_{pos}$** | **33.98** | **32.91** | **66.32** | **59.65** | **55.74** | **41.82** |

GRAPE$_{pos}$ outperforms advanced GCL methods in terms of clustering performance. This indicates that GRAPE$_{pos}$ explicitly brings a number of positives closer thereby the learned representations are more intra-class cohesive.

### F.2 On Heterophily Graph

**Table 5: Node classification accuracy in percentage with standard deviation on three heterophily graph datasets. The bold and underlined values indicate the best and the runner-up results respectively.**

| Methods | Chameleon | Squirrel | Actor |
|---|---|---|---|
| GCN | 63.90 ± 0.4 | 46.88 ± 0.4 | 28.33 ± 0.2 |
| GAT | 58.19 ± 0.4 | 41.66± 0.6 | 28.16 ± 0.2 |
| GAE | 47.94 ± 1.1 | 37.79 ± 1.0 | 25.18 ± 0.1 |
| VGAE | 44.46 ± 1.1 | 34.54 ± 0.7 | 25.68 ± 0.1 |
| DGI | 54.47 ± 0.6 | 40.96 ± 0.6 | 27.12 ± 0.1 |
| GMI | 49.05 ± 0.3 | 35.34 ± 0.3 | 25.90 ± 0.1 |
| MVGRL | 63.69 ± 0.5 | 49.14 ± 0.6 | 30.87 ± 0.0 |
| GRACE | 61.98 ± 0.4 | 38.83 ± 0.6 | 26.16 ± 0.0 |
| CCA-SSG | 60.01 ± 0.5 | 44.56 ± 0.5 | 28.15 ± 0.1 |
| BGRL | 63.28 ± 0.4 | 46.09 ± 0.3 | 28.24 ± 0.0 |
| ProGCL$_W$ | 62.10 ± 0.6 | 48.39 ± 0.6 | 28.42 ± 0.1 |
| COSTA$_{MV}$ | 60.27 ± 0.5 | 46.13 ± 0.3 | 28.55 ± 0.0 |
| **GRAPE$_{mask}$** | **66.58 ± 0.5** | 52.76 ± 0.4 | **33.89 ± 0.0** |
| **GRAPE$_{pos}$** | 62.20 ± 0.5 | **53.49 ± 0.3** | 30.68 ± 0.1 |

We further evaluate the performance of GRAPE on three heterophily graphs, with the results presented in Table 5. It is evident that

**Table 3: Statistics of datasets and corresponding hyperparameter settings.**

| Datasets | Domain | Nodes | Edges | Features | Classes | $\lambda$ | $\mu$ | $\tau$ | $K$ | $\rho$ | $intvl$ | $Epo$ |
|---|---|---|---|---|---|---|---|---|---|---|---|---|
| Cora | Citation Network | 2708 | 10556 | 1433 | 7 | 1 | 0.5 | 0.5 | 5 | 1 | 5 | 100 |
| CiteSeer | Citation Network | 3327 | 9228 | 3703 | 6 | 1 | 0.5 | 1.0 | 5 | 1 | 5 | 100 |
| PubMed | Citation Network | 19717 | 88651 | 500 | 3 | 1 | 0.5 | 1.0 | 4 | 1 | 5 | 100 |
| Wiki CS | Knowledge Base | 11701 | 216213 | 300 | 10 | 1 | 0.1 | 0.2 | 2 | 1 | 5 | 100 |
| Am-Photo | Social Network | 7650 | 119081 | 745 | 8 | 1 | 0.1 | 0.5 | 2 | 1 | 5 | 100 |
| Am-Computer | Social Network | 13752 | 245861 | 767 | 10 | 1 | 0.5 | 0.5 | 3 | 1 | 5 | 100 |
| Co-CS | Citation Network | 18333 | 81894 | 6805 | 15 | 1 | 0.5 | 0.5 | 3 | 1 | 5 | 100 |
| Co-Physics | Citation Network | 34493 | 247962 | 8415 | 5 | 100 | 0.5 | 0.5 | 2 | 1 | 5 | 100 |

GRAPE achieves superior clustering performance compared to advanced GCL methods. This superiority stems from GRAPE's ability to adapt well to heterophily graphs, as it does not rely on short-distance connections. It is worth noting that, although the nodes connected in heterophilic graphs are dissimilar, they generally adhere to label consistency across edges. Therefore, our initial and maximum ranges for the false negatives set based on hops are still reasonable.

### F.3 On Scalable Parameterization

In this subsection, we show the performance and efficiency under different strategies (shown in appendix B) of solving for the self-expression coefficients. Taking GRAPE$_{mask}$ as the example, the test accuracy and training time are exhibited in Table 6. It is evident that our method can strike a balance between efficiency and performance, making it suitable for large-scale graphs. Nevertheless, it's worth noting that scalable parameterization is still based on negative samples, and the memory overhead during training cannot be substantially reduced.

**Table 6: Test accuracy (%) and training time (s) on different parameterization.**

| Methods | PubMed | Am-Photo | Am-Computer |
|---|---|---|---|
| Full | 81.50 (116.59) | 93.32 (22.57) | 88.42 (66.32) |
| MLP | 79.26 (93.08) | 92.10 (18.61) | 87.59 (51.19) |
| Attentive | 79.43 (87.49) | 91.92 (18.03) | 87.50 (50.80) |

### F.4 Integration onto BGRL

As a negative hardness estimation scheme, GRAPE can be seamlessly extended to GCL methods, regardless of whether they are based on negatives or not. Considering a large-scale GCL method, BGRL [67], GRAPE$_{pos}$ can be directly incorporated into its loss function. The comparative results between the original BGRL and BGRL with GRAPE are presented in Table 7. There are consistent performance improvements achieved by the latter. Furthermore, these negatives-independent methods reduce memory consumption and thus are more suitable for scaling to large-scale graphs.

**Table 7: Test accuracy (%) of the extension to BGRL.**

| Methods | PubMed | Am-Photo | Am-Computer |
|---|---|---|---|
| BGRL | 80.05 | 92.95 | 88.19 |
| BGRL w/ GRAPE | 81.19 | 93.80 | 90.06 |

### F.5 Supplementary Parameter Analysis

In this subsection, we probe the sensitivity of the interval for updating $C$ (i.e., $intvl$) and the range of hard negatives (i.e., $K$).

The impact of adjusting $intvl$ on test set accuracy is depicted in Figure 12. The regions covered by bands represent the range of results obtained from 10 independent runs, and the points on the curves correspond to the averages. When $intvl$ exceeds the total number of epochs, it reverts to the GRACE approach. It is evident that as $intvl$ increases, there is an overall decline in accuracy. Consequently, frequent updates of self-expression coefficients entail additional computational overhead. Both $intvl = 5$ and $intvl = 10$ are deemed acceptable in our experiments.

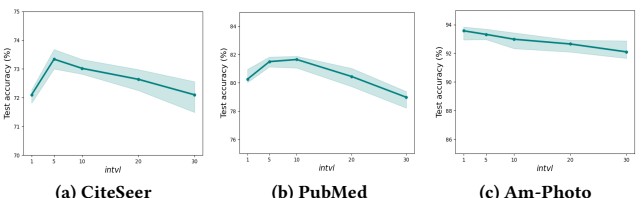

| (a) CiteSeer | (b) PubMed | (c) Am-Photo |
|---|---|---|

**Figure 12: Parameter analysis on $intvl$.**

The impact of hyperparameter $K$ on the results is illustrated in Figure 13. Due to variations in the inherent density of the graph datasets, the optimal $K$ value also varies. For sparse graphs, mining their expansive hard negatives indeed enhances performance but introduces the risk of subspace fallacy. In summary, GRAPE demonstrates stable performance in both narrow and expansive ranges of hard negative mining, validating the efficacy of our adaptive selection for hard negatives.

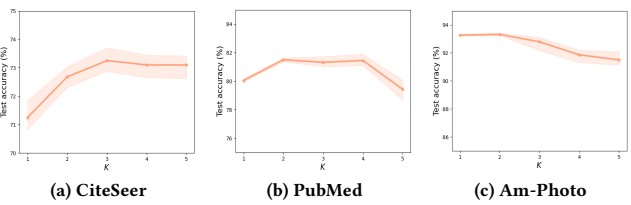

| (a) CiteSeer | (b) PubMed | (c) Am-Photo |
|---|---|---|

**Figure 13: Parameter analysis on $K$.**

Furthermore, in contrast to methods such as BGRL, we have observed that GRAPE displays a relatively low sensitivity to graph augmentation. All results are under the setting with 40% edge removal and 10% feature masking, which provides a substantial advantage compared to other GCL methods.

