# OpenReview forum: "Towards Expansive and Adaptive Hard Negative Mining: Graph Contrastive Learning via Subspace Preserving"
_ACM.org/TheWebConf/2024/Conference — TheWebConf24 Oral_

### Official Review · Reviewer_9Uus · 2023-11-21

**Novelty:** 6
**Technical Quality:** 6

**Review:**

Summary:
This paper presents a novel method, GRAPE, for Graph Contrastive Learning (GCL) through Subspace Preserving. The authors suggest that a more comprehensive and adaptive approach to hard negative mining could significantly enhance node-level GCL. GRAPE, a unique method for estimating negative hardness in GCL, is based on subspace theory. It can capture hard negatives beyond the scope of message passing and adaptively scale the set of hard negatives. To mitigate the impact of false negative samples on GCL, the authors have devised two strategies. Compared to several advanced GCL methods, GRAPE demonstrates superior performance across eight widely used public graph datasets. The paper also provides a theoretical explanation of GRAPE's properties and its relationship with related methods.

Strength Points:
S1. The authors provide comprehensive references to support their claims, justifying the motivation of this paper.
S2. Very solid technical details are provided.
S3. The proposed method, to some content, addresses the issues of hard negative sampling in the graph contrastive learning area.

Weakness Points:
W1. The beginning part of the introduction section introduces too much content about the background, which should be moved to the related work section. Too much unnecessary content may cover the core idea of the proposed method.
W2. The last paragraph of the introduction is disconnected from the previous paragraphs. The solution suddenly appears. The proposed solution should highly focus on the problems in existing research works.
W3. The organization is poor, especially for Section 3.2. It contains too much content in this subsection, failing to reveal a clear structure of the proposed method.

In short, the paper proposed a novel method utilizing hard negative samples based on subspace preserving and providing comprehensive details, showing the technical soundness of this paper. However, the presentation and organization are poor. The core ideas are flooded by those technical details. It is difficult for readers to extract the intuitions behind the proposed method.

**Questions:**

Q1. What is the subspace hypothesis in clustering and how does it relate to GRAPE? It would be better to give a more clear answer to justify the motivation to utilize subspace preserving.

**Reviewer Confidence:**

3: The reviewer is confident but not certain that the evaluation is correct

**Scope:**

4: The work is relevant to the Web and to the track, and is of broad interest to the community

---

### Official Review · Reviewer_Fz5q · 2023-11-24

**Novelty:** 6
**Technical Quality:** 5

**Review:**

This paper proposes to mine 'adaptive and expansive' hard positive/negative pairs for contrastive learning. The solution is inspired by 'subspace preserving': points in a subspace can be expressed as a linear combination of other points in the subspace. From this idea, the 'hard' samples are generated via an optimization that minimizes the difference between the anchor and hard negatives of the anchor.

Pro:
1. Identifying hard positive/negative pairs in graphs is a less explored area in graph algorithms, and this work gives a promising method.
2. The method is well-motivated.

Con:
1. Several key parameters and definitions are not well introduced, making the paper hard to read. (a). The paper spends half a page introducing GCN and contrastive learning, which are trivial for readers in the field; but does not spend much space introducing 'subspace perserving', the key novelty of the paper. The author directly jumps into the definition in line 391, making it confusing on what 'subspace perserving' means and how it can output the hard negative samples. In addition, in definition of line 391, parameter c is never defined and it's unclear how the matrix H is obtained. I have to go to the pseudocode section at the very end of the paper to understand the meaning and the derivation of each parameter.
2. From what I understand, the 'hard negatives' are obtained by solving eq(8) and eq(11). From eq(8), min|z-Hc| is minimized by setting H as the most similar set of nodes to z. The method can be problematic when the graph is sparse and large. When the graph is sparse, you are essentially pushing away nodes of the same classes because they are most similar to z. In addition, the hop distance will also increase in a sparse and large graph.
3. If you set H as the set of neighbor nodes of z, because z is obtained from GCN, isn't |z-Hc| = 0?
4. In evaluation, BGRL outperforms Grace in most datasets. Why don't you use BGRL as the basis to build your model? I understand you have a section in appendix, but you could use BGRL as the basis throughout the paper.

**Questions:**

See above for Q2-4.

**Ethics Review Description:**

N.A.

**Reviewer Confidence:**

3: The reviewer is confident but not certain that the evaluation is correct

**Scope:**

4: The work is relevant to the Web and to the track, and is of broad interest to the community

---

### Official Review · Reviewer_nS2u · 2023-11-24

**Novelty:** 5
**Technical Quality:** 5

**Review:**

This paper presents a method to improve the performance of current graph contrastive learning models. This paper points out that hard negative sampling techniques in GCL prefer to sample a large number of false negatives, pushing away the semantically similar representations. Therefore, this paper proposes a method named GRAPE, which aims to exclude the highly probable false negatives from the negative set. In practice, this paper develops two pluggable schemes, both excluding the highly probable false negatives from the negative set, to enhance GCL. Extensive experiments demonstrate the effectiveness of GRAPE.

> Strength:
- This paper conducts extensive experiments to evaluate the performance, time cost, visualization, influence of the hyperparameters, and so on, making GRAPE convincing enough.

- This paper provides detailed theoretical proofs and analysis of the theorems and propositions.

> Weaknesses:

- I think there are some blurred definitions in the equations. In Line 301, Φ_i is defined as a hard negative set, including both true and false negatives similar to the anchor according to the definition above. However, it seems that positive samples should be the anchor itself and the false negatives in Eq(5). Therefore, should Φ_i be defined as the false negatives set? The same applies to Eq(6).

- In Line 385, what exactly does “graph homophily” mean? Does it mean mining hard negatives within graph-structured neighbor sets?

- It’s not clear how to initialize the hard negative sets or determine the candidate sets of hard negatives before training.

- There is no clear description of how to select the false negatives and what “hard negative” and “false negatives” mean respectively.

**Questions:**

See weaknesses.

**Ethics Review Description:**

No ethics issue

**Reviewer Confidence:**

3: The reviewer is confident but not certain that the evaluation is correct

**Scope:**

3: The work is somewhat relevant to the Web and to the track, and is of narrow interest to a sub-community

---

### Official Review · Reviewer_ktgm · 2023-11-30

**Novelty:** 5
**Technical Quality:** 6

**Review:**

This paper focuses on the problem of mining negative samples in GCL, and is dedicated to exploring an effective mining strategy to find suitable hard negative samples, so as to improve the performance of GCL. The authors first conduct an empirical study of hard negative sample mining and conclude that more effective adaptive solutions. Then, it’s also combined with subspace preservation to solve this problem. The proposed method GRAPE achieves excellent performance on eight datasets. The authors further analyze the mining quality of negative samples and show visualization results.

Strengths
1. The article has a coherent logical flow, making it easy to read. Additionally, the illustrations in the text are aesthetically pleasing.
2. Negative sample mining is a classic issue, but there has been relatively limited exploration in the field of image contrastive learning. The method proposed by the author contributes to the development of this domain.
3. The proposed GRAPE demonstrated overall strong performance in the experiments.
4. The author provided theoretical justification for the proposed method from the perspective of mutual information.

Weaknesses
1. There are some detailed errors in the article, such as the spelling "massage" in Figure 1.
2. The article lacks a detailed analysis of the specific differences between the "mask" and "pos" strategies.
3. The method proposed in paper seems to be applicable only to homogeneous graphs.

**Questions:**

1. As mentioned above, some detailed errors should be corrected.
2. From the experimental results, the "mask" strategy outperforms the "pos" scheme in most of the datasets, can you give some explanations?
3. Can the proposed method be extended to heterogeneous graphs?

**Reviewer Confidence:**

4: The reviewer is certain that the evaluation is correct and very familiar with the relevant literature

**Scope:**

3: The work is somewhat relevant to the Web and to the track, and is of narrow interest to a sub-community

---

### Decision · Program_Chairs · 2024-01-22

**Decision:**

Accept (Oral)

**Comment:**

This paper is focusing on improving hard negative sampling for Graph Contrastive Learning (GCL), currently a costly task which if done well can allow the contrastive learning process to perform as intended. The paper introduces methods that can be used alongside a variety of GCL models and their goal is to improve our ability to identify negative samples for GCL. The proposed method is well-motivated and appears to work well on the experimental evaluation.

 Overall, the reviews recognize the novelty and importance of the proposed method and discussions address the raised concerns to a satisfactory degree.